# Interventions to Strengthen Environmental Sustainability of School Food Systems: Narrative Scoping Review

**DOI:** 10.3390/ijerph20115916

**Published:** 2023-05-23

**Authors:** Grace Gardner, Wendy Burton, Maddie Sinclair, Maria Bryant

**Affiliations:** 1Public Health Department, Newcastle City Council, Newcastle upon Tyne NE1 8QH, UK; grace.gardner@newcastle.gov.uk; 2Department of Health Sciences, University of York, York YO10 5DD, UK; maria.bryant@york.ac.uk; 3School of Health and Wellbeing, University of Glasgow, Glasgow G12 8QQ, UK; m.sinclair.3@research.gla.ac.uk; 4Hull York Medical School, University of York, York YO10 5DD, UK

**Keywords:** school food system, sustainability, planetary health, population health

## Abstract

School food systems play a role in the wider food system, but there is a scarcity of literature exploring interventions that aim to improve the environmental sustainability of school food systems. The present review aimed to understand and describe the types of interventions that have previously been explored to strengthen the sustainability of school food systems along with their impact. We applied a scoping review methodology guided by Arksey and O’Malley, which included a review of the online database Scopus and the grey literature. Information relating to intervention design, study population, evaluation method and impact were extracted. In total, 6016 records were screened for eligibility, 24 of which were eligible for inclusion. The most common types of interventions were school lunch menus designed to be more sustainable; school food waste reduction; sustainable food system education using school gardens; and dietary interventions with added environmental components. This review highlights a range of interventions which could positively influence the environmental sustainability of school food systems. Further research is needed to explore the effectiveness of such interventions.

## 1. Introduction

Globally, our food system contributes to at least 30% of all human-made greenhouse gas emissions and negatively impacts both planetary and population health [1]. Unsustainable food sources are a key contributing factor to this, including the mass production of animal-based products, food waste and food miles. In 2015, in an attempt to achieve a more sustainable future worldwide, 193 Member States of the United Nations adopted 17 Sustainable Development Goals (SDGs) as part of the 2030 Agenda for Sustainable Development [2]. These goals aim to combat issues of sustainability and are universal and ambitious in their plans. Examples of SDGs involving sustainability of the food system include the goal to end hunger, achieve food security and improved nutrition and promote sustainable agriculture; the goal to ensure inclusive and quality education for all and promote lifelong learning; and the goal to ensure sustainable consumption and production patterns [3].

School food systems play an important role in the overall food system. The school food environment contributes to the development of children’s dietary preferences and eating behaviours and therefore has the potential to play a meaningful role in the shift toward a more sustainable wider food system. Children spend a large proportion of their time at school, and an average of 30% of children’s daily energy intake is suggested to come from the school food [4]. Existing school food intervention studies have tended to focus on increasing children’s fruit and vegetable intake, improving the nutritional quality of food on offer or the food environment [5,6], with the aim of reducing health inequalities and incidence of diet-related disease [7]. However, few studies have explored interventions aiming to strengthen the environmental sustainability of school food systems and the wider impacts of this on the wider food system.

It is recognised that the production of food of animal origin has a great impact on the environment. The livestock sector contributes 14.5% of global greenhouse gas (GHG) emissions [8], with plant-based foods exhibiting lower environmental impacts than meat-based [9,10,11]. Red meat-based school meals have been shown to be major contributors to GHG and water consumption compared to other school meals [12,13]. In England, the carbon footprint from primary school meals produced over one year was estimated to be approximately 319 million kgCO_2_ equivalent, of which meat dishes were responsible for 52% [14]. Food waste generated by school food systems also contributes to the wider issue of food waste. Globally, our overall food waste is estimated to be one-third of all food produced [15]. This is echoed within school food systems, with one study estimating that 28.59% of the food prepared in Italian primary schools was not consumed by the diners [16], and another reporting that 23% of the food served in schools across Sweden was wasted [17]. Many factors have been identified as influencing food waste in schools, including the amount of food prepared by catering teams, serving size, eating environment and menu composition [18,19]. Therefore, a range of approaches aimed at varying stakeholders within the school food system (e.g., teachers, parents, caterers and the children themselves) may be required to address the problem, particularly as many schools engaged in environmental sustainability efforts may not be aware of how much food is wasted in their school [20]. While tackling the issue of food miles can be complex, due to the food mile concept often being oversimplified [21], other initiatives that may promote the environmental sustainability of school food systems include school gardens and food education programmes. School gardens provide an opportunity to teach children where their food comes from and how they could produce food themselves, thus potentially changing the behaviours of future generations, along with offering a potential local food source [22,23]. Schools are also being increasingly encouraged to purchase food from local and organic suppliers, such as farm-to-school programmes [24].

While anecdotal evidence suggests that some schools and communities are utilising these types of local initiatives to strengthen the sustainability and environmental impact of school food, there is still a need for more research in this area to understand their design, feasibility and potential impact. This scoping review was therefore conducted for the purpose of mapping and identifying the available evidence from research describing sustainable food system initiatives within the school context.

## 2. Materials and Methods

A scoping review methodology was applied to enable the existing literature to be explored broadly, to identify gaps in the research on sustainable school food systems and to allow exploration into how research has been conducted. Taking influence from Arksey and O’Malley [25], the methodology of this scoping review was conducted over four key stages: (1) identifying the research question; (2) identifying relevant studies; (3) study selection and (4) charting the data.

### 2.1. Identifying the Research Question

The research question was ‘Which types of interventions have been explored to strengthen the environmental sustainability of school food systems, and what was their impact?’. This was intended to help inform future research in the area of school food systems and facilitate a positive shift toward a more sustainable wider food system across the globe.

### 2.2. Identifying Relevant Studies

Our search strategy aimed to identify papers consistent with the concept of food and sustainability, which were undertaken in a school setting. We did not limit the search to a specific population in the school setting as we were interested in a broad range of stakeholders (e.g., children, caterers and teachers). The electronic database ‘Scopus’ was the chosen database to search for interventions on sustainable school food systems, as this has broad coverage across a vast number of disciplines. Other databases were searched during the development of our search strategy; however, they did not yield any extra papers of relevance. The grey literature was also searched using the same search terms, with the first ten Google Search pages being screened for eligible literature. Relevant papers that were identified using citations within included papers were also included in the review if they met eligibility criteria (as agreed between two members of the team: WB and GG), to ensure the inclusion of relevant studies that were not picked up within the original scoping strategy.

The identification of articles was carried out by searching the Scopus database using the following key search terms: (TITLE) (sustainab* OR “greenhouse gas” OR “climate change” OR “climate friendly” OR eco-school* OR food OR diet OR nutrition OR agri-food OR “food waste”) AND (TITLE) (school). The source type was limited to (Journal).

For the grey literature search, an advanced Google search with the exact same Scopus search criteria: (sustainability OR “greenhouse gas” OR “climate change” OR eco-school* OR food OR diet OR nutrition OR agri-food OR “food waste”) AND (school) was completed.

In addition, we used (sustainability OR “greenhouse gas” OR “climate change” OR eco-school* OR food OR diet OR nutrition OR agri-food OR “food waste”) AND (school lunch).

Scopus was searched twice, once in July 2021 and again in December 2022. The grey literature search was also conducted twice, once in May 2022 and once in December 2022. The most recent search in December 2022 was undertaken to ensure the scoping review was up to date, given the growing number of papers being published in recent years.

### 2.3. Study Selection

Our search strategy returned a large number of studies that fell beyond the scope of interventions to strengthen the sustainability of school food systems. This was due to broad search terms being included in the strategy (e.g., climate change, school, food). During the development of the search strategy, we identified that these broad terms were important for picking up records that represented a range of school food system interventions. However, in order for record screening to be manageable (as recommended by Arksey & Malley [25]) we developed eligibility criteria to help us eliminate studies that did not address our research question. Operationally, these criteria allowed us to identify records that were consistent with the following study concept, setting and evidence source. 

Concept: Studies were included if they described an intervention with the purpose of strengthening the environmental sustainability of a school food system. As our research objective was to also explore the design, delivery and impact of such interventions, the intervention design, delivery method and evaluation needed to be reported in all included papers.

Setting: As our research question aimed to explore interventions aimed at strengthening the environmental sustainability of school food systems, the study setting needed to be in a school context. Therefore, studies were included if the intervention was delivered in an early year, primary school or secondary school setting. Interventions could be aimed at any school stakeholder within the school food setting (e.g., pupils, teachers, catering staff) that were undertaken in the UK or a country that was comparable to the UK, so that potential for transferability could be considered (defined according to the World Bank List 2020/2021 https://datahelpdesk.worldbank.org/knowledgebase/articles/906519-world-bank-countryand-lending-groups: accessed 22 July 2022). 

Evidence source: Eligible sources were from peer-reviewed journals, as well as pre-specified grey literature sources. Google was used to identify papers, from which we accepted peer-reviewed publications as well as any article or report that described the evaluation of a sustainable school food system intervention. Included studies were those that were published in the English language or available as translated English versions. In order to be included, studies had to be (1) qualitative, quantitative or mixed methods evaluation studies; (2) systematic reviews or other reports/reviews that collate primary research; (3) case studies and/or (4) modelling studies. Studies were excluded if they did not describe the design, delivery method or evaluation of an intervention aimed at strengthening the sustainability of a school food system. Studies were also excluded if they were undertaken outside a early year, primary school or secondary school setting or described an intervention that was not based in the UK or a comparable country according to the World Bank List 2020/2021. Evidence sources were excluded if they were conference abstracts, theses/dissertations, discussion papers or book chapters, as these were judged to be difficult to read/manage within the timeframe and/or did not adequately describe primary research studies.

Studies were screened initially by title and abstract against the inclusion and exclusion criteria and were subsequently removed if they were not eligible. Initial screening was undertaken by three authors of this article (GG, WB and MS). All records were divided equally between reviewers to assess eligibility. To ensure consistency in the reviewer’s understanding and interpretation of the criteria, just 20% of each reviewer’s records were assessed for eligibility in the first instance. The same records were then assessed for eligibility by a second reviewer within the team. Following the second reviewer’s assessment of eligibility, the team met to discuss uncertainties and disagreements. Once all reviewers reached an agreement on eligibility, the remaining records were assessed. The review team continued to meet regularly throughout the review process to discuss and agree on the eligibility of any remaining records if not initially clear.

### 2.4. Charting the Data

We used a ‘narrative review’ charting approach to analyse the data as recommended by Arksey and O’Malley [25]. This approach involved extracting and collating standard information from each identified study to provide a comprehensive summary of the evidence, which allowed the research questions to be answered. Data relevant to answering the research questions were extracted and collated by the research team using an MS Excel spreadsheet including (1) publication date; (2) country where the intervention was delivered; (3) study population; (4) intervention components; (5) intervention duration; (6) study design(s); (7) comparator group(s); (8) outcome measure(s) related to sustainable school food systems and (9) impact/ results. These characteristics were summarised to describe the breadth of the data and then were categorised according to intervention type. Extracted data describing the study design, intervention and results of the evaluations within each category were summarised and mapped to provide an overview of the types of interventions that have been designed and evaluated with the aim of positively influencing a school food system and their reported impact.

## 3. Results

A total of 6016 records were screened for eligibility. Of these, 5845 were removed after the primary screening, and 171 full texts were assessed for eligibility. Four additional records were identified from the citations, resulting in 24 studies/reports being included in the review (Figure 1). The dates of publication ranged from 2011 to 2022. The majority of interventions were undertaken in Europe: Spain [26,27,28,29,30,31], Sweden [32,33,34,35], France [36,37], England [38,39], Finland [40] and Denmark [41]. Four were undertaken in the USA [42,43,44,45,46], one in Mexico [47] and one in Australia [48]. Seventeen of the studies used quantitative methods only to evaluate their intervention to promote the sustainability of the food system [26,27,28,29,32,34,35,36,37,38,39,40,41,43,44,45,46,49], four used qualitative methods only [33,42,47,50] and two used a mixed methods approach [30,48].

Of the studies which used quantitative or mixed methods (*n* = 20), a range of study designs was identified, the most common being a pre–post design without a control group comparator [34,35,38,44,45,46,48,49]. Five were modelling studies [26,27,28,29,36], three used a pre–post design with a comparator [30,32,40], two used a cross-over design [37,41], one used a historical control [39] and one used a cluster randomised trial design [43]. The studies that used qualitative methods only differed in their approach, with one undertaking focus groups only [33], one conducting semi-structured interviews only [50], one using an action research approach [47] and another using interviews, focus groups and observation with a case study design [42].

The types of interventions fell broadly into four categories: (1) school menus designed to be more sustainable, (2) food waste reduction, (3) sustainable food system education using school gardens and (4) dietary interventions with added environmental components. The characteristics of the studies are described in Table 1, Table 2, Table 3 and Table 4 with records arranged in descending order.

### 3.1. School Lunch Menus Designed to Be More Sustainable

Of the 24 studies identified in the review, 13 tested an intervention that aimed to promote the sustainability of school lunch menus. Of these, six explored the implementation of a sustainable lunch menu in schools [34,35,40,41,44,45], five used mathematical modelling techniques to simulate the environmental impact of menus optimised for sustainability [26,27,28,29,36], one used a qualitative study design to explore barriers and levers to implementing a more sustainable school menu [33] and one evaluated the impact of an intervention (Food for Life Partnership) on use of local suppliers [38].

The five modelling studies differed in their approach for optimising menus to promote their environmental sustainability. One optimised menu was underpinned with the ‘WEF-nexus’ approach, a concept that analyses the interactions between three environmental resources (water, primary energy and food systems) and identifies synergies and trade-offs between them [29]. Another was underpinned with a set of new agro-ecological policies that were planned to be shortly implemented [27], including a shift toward seasonal consumption, packaging reduction and green electricity. Two reduced the amount of meat on the menu amongst other scenarios (e.g., astringent menu, without fish, without eggs, fewer meal components) [28,36], and one balanced the carbon footprint of each item with nutrient value and cost [26]. All studies compared optimised menus with baseline menus. Three studies reported a >40% reduction in greenhouse emissions following menu optimisation [27,29,36] and two reported a >23% reduced carbon footprint [26,28]. Two studies also compared different types of optimised menu scenarios [28,36], identifying an astringent menu (menu designed to avoid causing stomach upsets using cooking techniques such as boiling and baking) [28], a menu without meat [28] and a menu with more vegetarian options [36] as having the lowest carbon footprint.

Of the six studies that measured the impact of implementing a menu designed to be more environmentally sustainable in schools, two tested menus optimised to reduce their carbon footprint [34,35], three tested a meat-free day [40,44,45], and one tested a traditional Nordic diet (comprising environmentally friendly and locally sourced hot foods). Elinder et al. [35,41] and Colombo et al. [34] used the same modelling approach to optimise their school lunch menus to be lower in greenhouse emissions. They assessed food consumption and food waste levels during a three-week delivery period in primary schools to assess how pupils responded to the menus. Both studies found no significant pre–post differences in food consumption or food waste overall, but Elinder et al. [35] did report that one school out of four had a significant pre–post increase in food waste. A follow up qualitative study undertaken by Colombo et al. [33] explored barriers and levers to implementation of their optimised menu. They held focus group discussions with kitchen staff and pupils, which revealed variations in how the menu was received, with some pupils not noticing a change whilst others noticed more vegetarian food. Some pupils expressed that food tasted better during the implementation period, although kitchen staff perceived there to be hesitance toward trying plant-based foods. Kitchen staff described challenges in working with the new menu, including time, budget, palatability and management of leftovers, but it was also considered it to be fun to try new recipes.

In the three studies testing meat-free days [40,44,45], the initiatives were already being implemented prior to commencement of the studies. The findings in these studies were mixed. Blondin et al. [45] and Hamerschlag et al. [44] measured the sustainability impact (including carbon footprint, water footprint, purchase of animal products and cost) of more sustainable menus compared with pre-intervention menus. Blondin et al. [45] reported no differences when considering the menu in its entirety (i.e., when including days where meat was consumed), though they did find a significant reduction in greenhouse emissions of 73.7% and a 50% reduction in water resources when comparing meat-free days with a pre-intervention, standard menu day. Hamerschlag et al. [44] reported a greenhouse gas emission reduction of 14%, reduced water footprint of 6% and a cost saving of USD 42,000 (nearly 1% less spent per meal) following a 30% reduction in the purchase of animal products. Lombardi et al. [40] measured participation in school lunch, food taken and food waste in a sample of schools where a meat-free day was being implemented using a pre–post design with a comparator group (intervention: *n* = 33 schools, control: *n* = 10 schools). They found no difference between groups fin any of the outcomes but did report a significant pre–post increase in food waste in intervention schools at the 11-week follow-up (35 g per participant vs. 56 g per participant) compared to control schools (30 g vs. 32 g). However, the authors reported that this levelled out by the 23-week follow-up. A traditional Nordic diet (comprising environmentally friendly and locally sourced hot foods) was tested in nine schools (*n* = 197 pupils) that previously only had a packed lunch option using a cluster-randomised controlled, unblinded cross-over design, which also found mixed results [41]. Food taken and food waste was compared between the traditional diet period and a packed lunch-only period. The results showed a higher amount of food taken during the traditional diet period compared to the packed lunch-only period, but there was more food waste.

### 3.2. Food Waste Reduction

Six of the studies identified in this review aimed to reduce the amount of food wasted from school lunches [30,32,37,46,48,49]. Two main types of interventions were explored: changes implemented within the dining environment (including changes in the way foods were served and environmental prompts) [31,32,37,46] and educational interventions delivered to children [30,48]. Of the four studies exploring changes in the dining environment, two used a pre–post design [32,46], one used a pre–post design with a comparator [32] and one used a cross-over trial design with six schools (*n* = 247 participants) [37]. The tested strategies included: offering both hot and cold vegetables [46], offering dips with cut raw vegetables, offering sliced or cut fruit [41], improving the lunchroom atmosphere [46], provision of tasting spoons [32], awareness campaign [32,49], plate waste tracker [32], attendance forecasting [32], a hunger traffic light prompt [40] and increasing the number of starters offered at lunch (i.e., entrée or first course) [37]. The two studies using a pre–post design reported a significant decrease in total waste (7.01% [46]; 41% [49]). Malefors et al. [32] used a pre–post design with a control group comparator (intervention: *n* = 7 schools; control: *n* = 7 school) and reported that only the awareness campaign and attendance forecasting achieved a greater reduction in food waste than the control group, which had a 38% pre–post reduction in food waste. The cross-over trial by Rigal et al. [37] reported an increase in waste when offering more starters at lunch in six high schools in their cross-over trial with repeated measures.

Both studies testing an educational intervention to reduce food waste reported positive results. The two interventions differed slightly in length and focus. One was a classroom-based intervention delivered over three weeks, which focused on the social and environmental consequences of food waste [30]. The other was a classroom intervention delivered over six weeks, which focused on reducing waste associated with packed lunches, and included a parent component, encouraging them to involve children in the preparation of packed lunches to reduce waste [48]. Both studies reported a pre–post reduction in food waste. Anton-Peset et al. [30] also used two comparator groups (intervention group (one primary school class), *n* = 15 pupils; nursery, *n* = 48 pupils; rest of school, *n* = 100 pupils) to test their intervention focusing on social and environmental consequences. Their results showed that the intervention group had the greatest reduction in food waste (14.88%) compared with a 0.91% reduction in the nursery group and a 3.18% reduction in the rest of the school group. Boulet et al. [48] used a pre–post design with no comparator group to test their intervention with a focus on packed lunches, reporting a 35% waste reduction.

### 3.3. Food System Education using School Gardens

Three articles identified in this review described using qualitative methods to explore the experiences of teachers and volunteers who were involved in the delivery of school garden initiatives aimed at promoting sustainable school food systems [42,47,50]. All of the school garden initiatives differed in their approach; one involved primary and secondary school pupils from different parts of the world engaging in virtual exchanges about their school gardens [50], another involved garden lessons being delivered to pupils from a one-acre garden space [42] and the third involved teachers being trained on agroecological practices and biocultural heritage, aiming to influence their teaching practice [47]. All participants perceived the garden initiatives to be positive for promoting the environmental sustainability of the school food system, for example, a perceived increase in pupil awareness around food production and horticultural competencies [50] and described moments of reconnection happening constantly between pupils as both a producer and consumer of food [42]. Teachers themselves reported a greater understanding of sustainability concepts [47], but some challenges were described, including perceived stereotypes, norms and othering between different learning groups [50], a contrast between the outdoor, experiential-based learning of school gardens and the rigid structure of the standard school curriculum [42] and teachers failing to implement key concepts taught in the programme in their teaching practice [47].

### 3.4. Adding Environmental Messaging to Existing Dietary Interventions

Two studies built upon existing dietary interventions by adding on environmental concepts to promote engagement [39,43], and these studies reported mixed results. The primary outcome for both studies was fruit and vegetable intake, but both included a secondary outcome relating to sustainable food systems: attitudes towards sustainable food [39] and use of single-use packaging in packed lunches [43]. Jones et al. [39] added optional environmental components to a programme aiming to promote a whole-school approach to food, whereby schools are given a selection of resources and awarded for meeting food related objectives (Food for Life Partnership). Primary schools were given the choice of which additional components they wanted to implement in their school during the study, including sustainable food education, staff training on cooking and growing food and parent engagement strategies. They used a historical control design to measure changes in pupil attitudes toward sustainable foods following the implementation of environmental components, and they reported that more pupils had a positive attitude at stage two (18–24 months after school enrolment in the programme) compared to stage one (point of school enrolment in the programme). Goldberg et al. [43] added messages about the value of environmentally sound nutrition practices (including reducing the use of single-use packaging) to a standard classroom-based nutrition education intervention delivered over 22 lessons, but they found no difference between groups for single packaging use in their cluster randomised controlled trial.

## 4. Discussion

To our knowledge this is the first narrative scoping review to gather available research on interventions that aimed to strengthen the environmental sustainability of school food systems. The available sourced evidence focuses on four main areas of intervention: development of school lunch menus designed to be more sustainable, school food waste reduction, use of school gardens to promote food system education and adding environmental messaging to existing dietary interventions. The results of this review enable us to learn which types of interventions may have potential to strengthen the environmental sustainability of school food systems as well as offering direction for future research.

The majority of studies identified in this review explored the impact of optimising menus to be more sustainable. Many of these were modelling studies, which aimed to develop school lunch menus with a reduced environmental impact, and all reported improvements in environmental sustainability. A promising feature of many of the studies using this approach is that the menus had been implemented into a real-world setting, using routinely available menu or audit data [38,40,44,45]. This suggests the feasibility of the approach and that action is already being taken to reduce the environmental impact of school food systems in many areas. Outside of the school setting, others have used mathematical modelling techniques to understand the environmental impact of existing menus and developed tools to promote environmental sustainability. Sherry and Tivona [51] used a Life Cycle Assessment to determine the environmental impact of food purchased in a small college in the USA. Using this analysis, they produced a decision-making tool providing information on swaps that could be implemented by catering teams to reduce the environmental impact of their menu. In a different study, Brink et al. [52] used mathematical modelling techniques to produce population-level dietary guidelines in the Netherlands, which were optimised to strengthen environmental sustainability, again suggesting that this type of intervention is already being implemented on a wide scale; however, no data on acceptability was reported. As revealed by Colombo et al.’s [33] qualitative study, potential barriers could exist to implementing a more sustainable menu in a school setting, including pushback from kitchen staff and pupils. In a study exploring the willingness of parents to support a more sustainable school food menu in a school in Italy [53], the authors reported that most parents were not willing to pay extra for more sustainable school menus and were pessimistic about their children’s willingness to accept more environmentally sustainable foods.

Six of the studies identified in this review tested an intervention aimed at reducing food waste in schools. Two of the studies reporting a pre–post reduction in waste tested the impact of an educational intervention [30,48]. Fraj-Andre et al. [54] tested a similar approach in a higher education setting, whereby food waste education was provided in University student’s marketing subject modules in the USA. They reported pre–post changes in the student’s food waste behaviour and an increase in food waste concern. However, despite the indication of success using this approach, there remains a lack of definitive evidence for the effectiveness of food waste reduction interventions. Many studies identified in this review used a pre–post design; therefore, it cannot be concluded whether the reductions in food waste happened by chance or due to the engagement in the research. Moreover, none of the food waste interventions identified in this review applied a predefined target to define the impact of their intervention, potentially due to insufficient information on what level of reduction could be considered meaningful. Outside of an educational setting, Stöckli et al. [55] undertook a systematic review to understand the available evidence on consumer-level food waste reduction interventions. They also noted a lack of evidence for the effectiveness, acknowledging conceptual and methodological challenges to evaluating such interventions and recommending that standardised definitions and measurement methods should be used in future research. Moving forward, interventions defining what a meaningful target for school food waste is and using a rigorous evaluation design could help understand the extent to which school-based food waste interventions might have a positive impact on the sustainability of school food systems and beyond.

This review identified three studies which explored the experiences of teachers and volunteers engaged in school garden initiatives. All of the school garden interventions appeared to have some potential to positively influence school food systems, particularly the perceived engagement of children, which in turn could impact on their awareness of sustainable food issues. However, none of the studies gathered data from the pupils themselves in terms of how they experienced the programme. Quantitative data on behavioural and environmental outcomes of these interventions is also lacking in the literature, although there is a broad literature on school garden initiatives without an environmental sustainability focus. For example, the findings of a systematic review undertaken by Chan et al. [56] suggest that school gardens may be effective in promoting school children’s nutritional knowledge, attitudes and acceptability towards vegetables. Future studies should consider including an environmental outcome measure within the evaluation of school garden initiatives.

Two studies aimed to enhance existing school-based dietary interventions by adding messaging to outline the environmental benefits that can be achieved by eating more fruit and vegetables. Although the impact of this approach was not demonstrated in this scoping review, incorporating environmental messaging into the school curriculum, in subjects such as Geography and Science, to promote climate change literacy is an approach that has been successfully adopted previously [57,58,59]. Therefore, school curriculum-based interventions with a focus on food system literacy may be feasible.

### Strengths and Limitations

A key strength of this review is the fact it explored interventions aiming to strengthen the environmental sustainability of school food systems, which is a timely and important area of interest. This review also included a wide range of study designs, so it was possible to explore in-depth qualitative data as well as the available quantitative data. However, we acknowledge that there are a number of limitations regarding this review. Firstly, we decided to exclude interventions from this review which did not specify their aim of improving the environmental sustainability of school food systems. It is understood that some interventions, such as school gardens being designed to improve knowledge around food production, may result in indirect environmental benefits. Nevertheless, without environmental aims underpinning their design, causal pathways would not be clear. Secondly, within the present scoping review, only one database was searched, which was Scopus. This database was judged to be the most appropriate because it is a large database, which we found to cover the largest number of recent and relevant citations for the purpose of this review during our review design process. Further, we explored the degree to which using an additional database resulted in further papers and did not find that this was warranted in this scoping review (non-appreciable difference in the number of overall papers identified). We are aware that our scoping review methodology did not identify interventions that may have been tested outside of academia (e.g., school-led interventions) or that have not yet been formally evaluated. As such, the literature regarding the evaluation of these interventions may not be available. We would expect this to change in the coming years given the increased emphasis on combating the negative effects of climate change and promoting the sustainability of food systems.

## 5. Conclusions

There is still work that needs to be performed to strengthen the environmental sustainability of school food systems across the world. This review highlights key areas that could be built upon, which were shown to be successful on a small scale. These interventions could have the potential to positively impact the wider food system, if scaled up. The majority of papers published in this area were published in the last five years, emphasising the increasing interest and growth of research around environmentally sustainable school food. There are various implications for future research or practice that have emerged from this scoping review. Overall, there needs to be agreement on how to measure the impact of interventions aiming to promote the environmental sustainability of school food systems. In terms of study design, there is a need for more controlled studies on effectiveness to tease out the longer-term impacts against comparator schools and to disentangle the potential impact of being involved in research. For research on school gardens, there is a need for focus on the environmental impact of these and the potential success of integrating them within the wider school curriculum.

Currently, many school-based interventions focus on dietary health. However, the addition of initiatives aiming to improve planetary health in schools, alongside dietary initiatives, may have the potential to shape future ‘norms’ of food behaviours, encouraging children to consider what is best for the individual as well as the environment around them.

## Figures and Tables

**Figure 1 ijerph-20-05916-f001:**
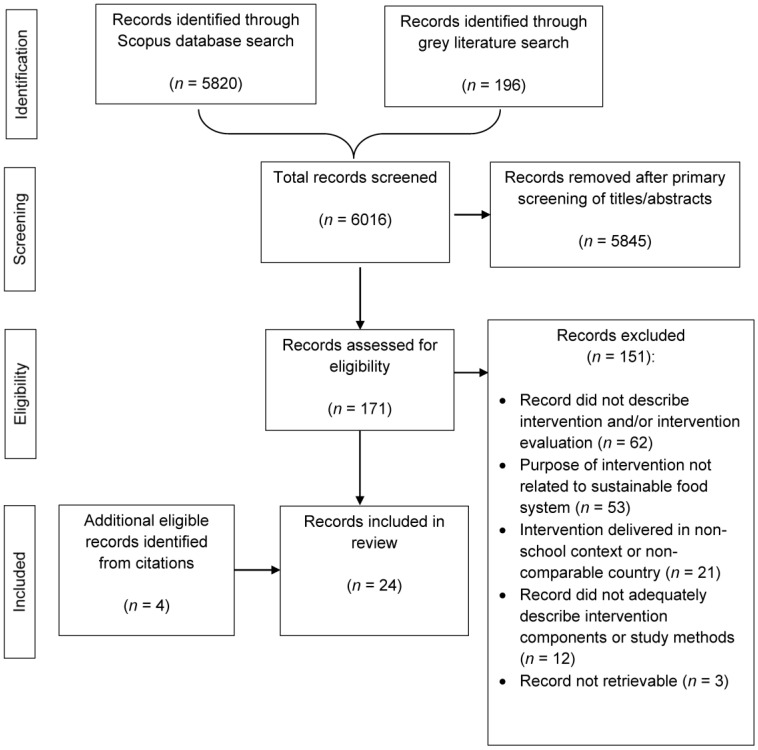
Scoping review search results.

**Table 1 ijerph-20-05916-t001:** Characteristics of interventions included in this review, where school lunch menus were designed to be more sustainable.

Author, Year and Location	Intervention Category	Sample Characteristics/Data Source	Intervention	Intervention Duration	Evaluation Design	Intervention Group	Comparison or Control Group	Outcome(s) Related to Sustainable School Food Systems	Main Findings
1. Poinsot et al., 2022 [36] France	School lunch menu designed to be more sustainable	Primary school lunch menus from schools in France where school meals are required to be made up of four or five components (i.e., a starter and/or dessert, a protein dish, a side dish and a dairy product).	School lunch menu optimised by considering four ‘trade-offs’: (1) Reducing the number of meal components; (2) Compliance with national school nutrition guidelines; (3) Increasing the number of vegetarian meals; (4) Avoiding ruminant meat.	N/A	Pre–post (Modelling study)	Optimised menu (four trade-offs compared)	Standard menu	Greenhouse gas emissions (% reduction in kg CO_2_ eq per meal)	Best pre–post reduction from more vegetarian meals (25% reduction).
2. Colombo et al., 2021 [33] Sweden	School lunch menu designed to be more sustainable	Primary school kitchen staff and pupils (aged 10–15) from schools in Sweden where the same lunch menu is provided to all schools, but each school chef has some degree of freedom to adapt menus to match preferences of their pupils.	Implementation of menu optimised to be 40% lower in greenhouse gas emissions.	4 weeks	Qualitative study: Focus groups (*n* = 9)	29 primary school children and 13 kitchen staff	N/A	Barriers and levers to successful implementation of sustainable men	Experiences with the menu: -Variations in how it was received; -A challenging experience to work with the new menu. The meaning of sustainability: -A broad and varied understanding of diet and sustainability; -Diet sustainability important but hard to realise. Plant-based acceptance: -Decisive role of taste, appearance, smell and recognition; -Habits, peer pressure and fears challenging acceptance. Opportunities to increase plant-based eating: -Focusing on familiar foods; -Increasing exposure, normalisation and motivation. -Gradual and realistic changes: Need for supportive environment: -More knowledge, resources and inspiration; -Increased stakeholder involvement.
3. Perez-Neira et al., 2021 [27] Spain	School lunch menu designed to be more sustainable	School lunch menu data from pre-schools and primary schools in Spain. Lunch menus developed by the local government school canteen network, with catering service provided by local kitchens.	School lunch menu optimised to simulate compliance with new agro-ecological policies on: (1) How products are produced; (2) Where products are produced and consumed; (3) When and how the products are consumed; (4) What products are consumed.	N/A	Pre–post (Modelling study)	Optimised menus	Baseline menus	Total GHG emission (% reduction in kg of CO_2_-eq per meal)	Pre: 1.36 kg of CO_2_-eq per meal. Post: 13.4% reduction if current trajectory followed but could rise to 40.6% if transformation advanced.
4. Batlle-Bayer et al., 2021 [29] Spain	School lunch menu designed to be more sustainable	High school lunch menus from schools in Spain, where meals consisted of two courses, dessert and bread.	School lunch menus optimised to simulate transition to low carbon meals using Nexus approach—considering following measures: (1) Blue water footprint (BWF); (2) Primary energy demand (PED); (3) Land use (LU); (4) Global warming potential (GWP).	N/A	Pre–post (Modelling study)	Optimised menu	Standard menu	% reduction in environmental impact (based on Nexus approach measures)	Optimised menu had the following reductions: 60% BWP, 46% PED, 48% LU and 53% GWP.
5. Colombo et al., 2020 [34] Sweden	School lunch menu designed to be more sustainable	Primary schools in Sweden, where children had two daily meals to choose from.	Implementation of menu optimised to be 40% lower in greenhouse gas emissions.	4 weeks	Pre–post	3 schools (*n* = 1635 pupils)	No comparison group	(1) Food waste (g/pupil); (2) Consumption (g/pupil); (3) School meal satisfaction (pre-post questionnaire).	(1) No pre–post difference in any of the participating schools; (2) No pre–post difference in any of the participating schools; (3) No pre–post difference in any of the participating schools.
6. Blondin et al., 2022 [45] USA	School lunch menu designed to be more sustainable	Schools from large urban school district in the USA, where one entrée was offered per day.	Meatless Mondays	One menu cycle (2–4 weeks)	Pre–post	One school district	No comparison group	(1) GHG emissions kg CO_2_-eq (per entree offered on a Monday and per entree averaged over week); (2) Water resources (litres).	(1) Significant reduction for pre–post meals offered on a Monday (0.95 vs. 0.25 kg CO_2_-eq), but no pre–post difference for meals averaged over week; (2) No significant differences.
7. Elinder et al., 2020 [35] Sweden	School lunch menu designed to be more sustainable	Primary school pupils from schools in Sweden serving a four-week menu plan (including 2–3 dishes/day over a period of 20 weekdays).	Implementation of a menu optimised to be 28% lower in greenhouse gas emissions.	4 weeks	Pre–post	4 primary schools (each with 360–660 pupils)	No comparator group	(1) Food consumption (g/pupil); (2) Food waste (g/pupil).	(1) No significant changes; (2) Plate waste significantly increased in one school (16 g/pupil to 21 g/pupil), but no significant changes overall.
8. Martinez et al., 2020 [28] Spain	School lunch menu designed to be more sustainable	Primary school lunch menus from schools in Spain designed following Spanish schools’ dietary guidelines.	School lunch menus optimised to reduce their carbon footprint considering food production, transportation and cooking. Six scenarios were considered: (1) Without dairy and legumes; (2) Without meat; (3) Without fish; (4) Without eggs; (5) Hypocaloric menu; (6) Astringent menu (menu designed to avoid causing stomach upsets using cooking techniques such as boiling and baking (e.g., boiled vegetables and chicken breast)).	N/A	Pre–post (Modelling study)	Optimised menu	Standard menu	Carbon footprint (kg CO_2_ eq.person/monthly).	Pre: 24.39 kg CO_2_ eq.person/monthly. Post: Greatest reductions from astringent menu (14.77 kg CO_2_ eq.person/monthly) and menu without meat (17.11 kg CO_2_ eq.person/monthly).
9. Hamerschlag & Kraus-Polk 2017 [44] USA	School lunch menu designed to be more sustainable	Primary, middle and high schools in the USA.	Climate-conscious menus implemented over one school district. The series of initiatives included: Meatless Monday, Lean and Green Wednesday and ‘California Thursdays’.	One season	Pre–post	85 schools (*n* = 37,000 pupils)	No comparator group	(1) Reduction in meat/dairy (lb per meal/%); (2) Greenhouse gas emissions (kg CO_2_-eq per meal served); (3) Water footprint (gallons per meal); (4) Cost saving ($/%).	(1) Pre: 0.14 lb; post: 0.10 lb per meal/30% reduction); (2) Pre: 0.70 kg CO_2_-eq per meal served; post: 0.61 kg CO_2_-eq per meal served; (3) Pre: 113 gallon; post: 106 gallon per meal served (4) USD 42,000 less spent per meal (1% per meal less).
10. Ribal et al., 2016 [26] Spain	School lunch menu designed to be more sustainable	School lunch menu data from one school catering company in Spain offering a large variety of meal combinations served with bread and water.	Optimisation of a menu that minimised cost and carbon footprint levels and promoted micronutrients.	N/A	Pre–post (Modelling study)	Optimised menu	Standard menu	Carbon footprint (kg CO_2_ equivalent)	A 23–24% reduction in the carbon footprint, but when balanced with the average budget, the reduction was 15–16%. Optimised menu had lower calcium content (below the set threshold), but the micronutrient energy share was more balanced.
11. Thorsen et al., 2015 [41] Denmark	School lunch menu designed to be more sustainable	Third and fourth grade primary school children from schools in Denmark with a previously packed lunch option only.	Traditional Nordic diet (environmentally friendly and sourced from the Nordic region).	3 months	Cluster randomised controlled unblinded cross over study	Pupils from 9 schools (*n* = 187)	Traditional Nordic diet vs. packed lunches	(1) Food intake (g); (2) Edible waste (g/%).	(1) Traditional Nordic diet: 230 g vs. packed lunch: 208 g; (2) Traditional Nordic diet: 88 g/29% vs. packed lunch: 43 g/16%.
12. Lombardini et al., 2013 [40] Finland	School lunch menu designed to be more sustainable	Primary and secondary school pupils from schools in Finland implementing a vegetarian day.	Weekly vegetarian day where no meat or fish products are offered (forced restriction) on the school lunch menu for one day each week.	11 months	Pre–post with comparator	33 schools	10 schools	(1) Participation in school lunch (%); (2) Food taken (g); (3) Food waste (g).	(1) Intervention: pre 83%; post 77%. Control: pre 78%; post 89%. No significant difference between groups; (2) Intervention: pre 288 g; post 35 g. Control: pre 333 g; post 316 g. no s.d. between groups; (3) Intervention: 35 g pre; 56 g post (significant pre–post reduction). Control group: pre 30 g; post 32 g. No significant difference between groups.
13. Orme et al., 2010 [38] England	School lunch menu designed to be more sustainable	Primary, secondary and special schools in England engaged in the Food for Life Partnership scheme.	Food for Life Partnership scheme (FFLP) including the following menu objectives: (1) Use of seasonal menus and in-season produce; (2) Display information about the origins of all fresh produce used; (3) Have at least 30% of ingredients from organic sources; (4) Have at least 50% of ingredients from local suppliers.	18 months	Pre–post	38 schools	No comparator	Number of schools using local suppliers (%)	Increase of 73% of schools using local suppliers.

**Table 2 ijerph-20-05916-t002:** Characteristics of school food waste reduction interventions included in this review.

Author, Year and Location	Intervention Category	Sample Characteristics/Data Source	Intervention	Intervention Duration	Evaluation Design	Intervention Group	Comparison or Control Group	Outcome(s) Related to Sustainable School Food Systems	Main Findings
1. Boulet et al., 2022 [48] Australia	Food waste reduction	Primary school children aged 5–12 and their parents from Australia in schools where students typically bring food from home.	Educational intervention for children and parents encouraged to involve children in packed lunch preparation to avoid waste. The intervention comprised: lessons for students, parent information and lunchbox ideas, hands-on workshop and ‘make your own lunch’ day.	6 weeks	Pre–post	Pupils (*n* = 775) and their parents from five schools (*n* = 4)	No control group	(1) Food waste (overall number of avoidable food waste items in packed lunch); (2) Self report eating of ‘all’ food at school (%); (3) Parental attitudes (qualitative methods).	(1) Pre: 218 avoidable items; Post: 141 avoidable items post; Self-report eating of all food at school. (2) Pre: 57.3% eating ‘all’ food; Post: 63% eating ‘all’ food; (not significant). (3) Greater interest and involvement of children in choosing and making food to take to schools—parents paid more attention to what they were providing to their children.
2. Vidal-Mones et al., 2022 [31] Spain	Food waste reduction	Pupils aged 3–18 from schools in Spain where students are in charge of setting the table and tidying it when they finished eating. All schools served three courses: first course (vegetables, pasta rice or legumes), second course (protein + salad, vegetable sauces or potatoes) and dessert (fruit or dairy product).	Three nudging strategies were designed and implemented: (1) letting students know the menu on the day before lunchtime (for cases 1 and 2); (2) making students reflect on their hunger level (for cases 1, 3 and 4) and (3) teaching students how to properly cut and eat fruits (for cases 1 and 2).	10 days	Pre–post	5 schools	No control group	Total food waste (kg)	Pre: 20.58 kg across all schools Post: 13.27 kg across all schools (significant reduction of 41%)
3. Malefors et al., 2022 [32] Sweden	Food waste reduction	Pupils aged 6–19 from schools in Sweden where food is served by a public catering organisation.	Four food waste strategies selected by public catering managers were tested: (1) Information campaign directed at school children; (2) Tasting spoons in canteens; (3) Plate waste tracker providing live feedback on how much food has been wasted; (4) Forecasting for canteens to help gauge attendance.	7 weeks	Pre–post with comparator	8 schools	Reference group (*n* = 7 schools)	Food waste for each strategy tested (g)	Awareness campaign: pre: 37 g; post: 24 g (significant reduction of 35%). Tasting spoons: pre: 27 g; post: 21 g (significant reduction of 22%). Plate waste tracker: pre: 19 g baseline; post: 12 g (37% reduction but not significant). Forecasting: pre: 69 g; post: 35 g (significant reduction of 49%). Reference group: pre: 58 g; post: 41 g (significant reduction of 38%). Only the awareness campaign and forecasting achieved greater plate waste reduction than the reference group.
4. Rigal et al., 2022 [37] France	Food waste reduction	Pupils aged 15–19 from schools in France with on-site cooking facilities, offering three or six starters in a self-service format, in which students serve themselves freely.	Different number of starters (first course of the meal or “entree”) offered at school lunch (three vs. six) at two time points (T1 and T2) to see which resulted in the most food waste.	School lunches offered at two time points	Cross-over trial with repeated measures: T1 (baseline), T2 (T1 + 21 days)	Pupils from six senior high schools (*n* = 247 pupils)	3 vs. 6 options of starter	Food waste (g)	Three starters: 47.58 g ± 7.35. Six starters 75.68 g ± 9.52. Increase of 28.10 g.
5. Anton-Peset et al., 2021 [30] Spain	Food waste reduction	Primary school children from one school in Spain with a mid-morning snack brought from home and lunch managed by a catering company.	Forty-five-minute teaching sessions including fifteen activities carried out to train pupils on the food waste concept and inform them about its impact.	3 weeks	Pre–post with comparator	Pupils from one primary school class (*n* = 15)	Nursery (*n* = 48 children) and the rest of the school (*n* = 100 children)	(1) Food waste (g/%); (2) Knowledge and attitudes: pre–post survey and qualitative methods.	(1) Intervention group: pre; 177 g/47.83% post; 101 g/32.95% post (reduction of 14.88%). Nursery group: pre; 87 g/23.30% post; 81/26.48% (reduction of 3.18%). Rest of school: Pre; 164 g/43.76% post; 130 g/42.85 post (reduction of 0.91%). (2) Knowledge and attitudes: subtle pre–post changes including an increase in identification of food waste concepts.
6. Elnakib et al., 2021 [46] USA	Food waste reduction	Primary and middle school children in the USA where meals are provided on-site.	Lunch time staff trained on how to reduce food waste in schools. Lunch time staff then selected strategies to be tested in their respective schools. Strategies included: offering both hot and cold vegetables, offering dips with cut raw vegetables, offering sliced or cut fruit and improving the lunchroom atmosphere.	4 weeks	Pre–post	Pupils from 15 schools	No control group	(1) Number of strategies implemented in each school (Mean and range); (2) Food waste (%).	(1) Number of strategies: Mean: 7.40 ± 6.97 SD; Range: 0 to 28 delivered consistently in each school; (2) Food waste: Significant pre–post reduction of 7.01% (β = −7.061, *p* < 0.001).

**Table 3 ijerph-20-05916-t003:** Characteristics of interventions included in this review, where sustainable food system education is provided using school gardens.

Author, Year and Location	Intervention Category	Sample Characteristics/Data Source	Intervention	Intervention Duration	Evaluation Design	Intervention Group	Comparison or Control Group	Outcome(s) Related to Sustainable School Food Systems	Main Findings
1. Lochner et al., 2021 [50] Germany	School gardens	Primary and secondary school pupils engaged in 16 Virtual school garden exchanges originating in England, Germany, India, Uganda, Mexico, Kenya, USA, Greece and Argentina.	Virtual school garden exchanges—primary and secondary school students from different parts of the world who work in school gardens engage in Virtual Exchanges (VEs) about their gardens and related topics. They use media such as photos, films and videoconferences	Ongoing	Semi structured interviews	24 educators from 9 different countries (England, Germany, Greece, Kenya, Uganda, Argentina, Mexico, USA and India) spanning 5 continents	N/A	Perceived learning outcomes of VGCE.	Perceived increase in pupil awareness around food production, climate, seasons, weather and eating habits and an increase in horticultural competencies such as gardening in greenhouses, keeping chickens, diversification and intercropping, dealing with pests, irrigation and composting. Data also revealed perceived stereotypes, norms and othering between pupils.
2. Ferguson et al., 2019 [47] Mexico	School gardens	Teachers from pre-, primary and secondary schools in Mexico.	In total, 120 h of teacher training to promote an understanding of agro-ecology. Modules include scientific process and thinking; health and nutrition; embracing local agro-ecological knowledge and foodways; strategies for garden program sustainability and design and application of garden-based lessons.	2 weeks	Qualitative active research (survey, self-evaluation, reflection, journals and interviews).	38 educators	N/A	Understanding of concepts described in the training and whether they influenced teaching practice.	Teachers reported greater understanding of key principles and essence of agro-ecology. Just over half identified one or more principles when asked to explain what they had learnt, whereas others did not identify a key principle but appeared to understand the essence of agro-ecology. In addition, only half of the participants attempted to use the learning in their teaching.
3. Cramer et al., 2019 [42] USA	School garden	School children from schools in the USA offering garden lessons.	Rural garden-based learning programme delivered from one-acre garden space (at least six lessons per year).	Ongoing	Qualitative case study (interviews, focus groups and observations)	Educators and founders of the programme (*n* = 8)	N/A	Perceived efficacy of school garden programme for food system ‘reskilling’.	Participants described, “moments of reconnection” happening constantly between students as both producers and consumers in the modern food system while students participated in planting seeds, tending crops and harvesting and sampling the fruits of their efforts. Garden educators also described feeling, “called to make the food system better”. However, barriers were expressed in terms of a contrast between the outdoor, experiential-based learning and the rigid structure of the standard school curriculum.

**Table 4 ijerph-20-05916-t004:** Characteristics of interventions included this review, where sustainable environmental components were added to dietary interventions.

Author, Year and Location	Intervention Category	Sample Characteristics/Data Source	Intervention	Intervention Duration	Evaluation Design	Intervention Group	Comparison or Control Group	Outcome(s) Related to Sustainable School Food Systems	Main Findings
1. Goldberg et al., 2015 [43] USA	Dietary intervention with environmental components	Third and fourth grade pupils from schools in the USA who brought food from home at least three times per week.	Great taste less waste (GTLW): Standard nutrition education delivered in 30 min classroom lessons with added environmental components including: campaign kits with reusable food containers and a packaging guide with information about purchasing and packing healthy lunches. Monthly parent newsletter sent home with nutrition advice and seasonal recipes.	22 lessons	Cluster RCT	5 schools (*n* = 327 children)	Standard nutrition intervention (2 schools; *n* = 78 children) and control group (5 schools; *n* = 177 children)	Mean prevalence of single use packaging (%).	GTLW: 57.4%; Standard nutrition intervention: 61.7%; Control: 60.4%; No significant difference between groups.
2. Jones et al., 2012 [39] England	Dietary intervention with environmental components	Primary school pupils from schools in England engaged in the Food for Life Partnership scheme.	Additional components added to the existing whole school approach initiative (Food for Life Programme) to incorporate sustainable food issues including food quality and procurement, food education and parental and community involvement, with the aim of promoting fruit and vegetable intake. Schools selected their own strategies for implementation	18–24 months	Historical control	Stage 1 (point of enrolment with FFLP): 1435 pupils	Stage 2 (18–24 months after enrolment): 1463 pupils	Positive attitude towards sustainable food (%).	Stage 1: 10.7%; Stage 2: 21.8%.

## Data Availability

Not applicable.

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
