# Peer review of "Interventions to Strengthen Environmental Sustainability of School Food Systems: Narrative Scoping Review"

_ijerph, 2023, doi:10.3390/ijerph20115916_

Round 1
Reviewer 1 Report
The subject matter of the reviewed article concerns sustainable school food systems. This topic is extremely relevant given the context of the triad: economy-society-environment. The essence of the article is a review of the literature in this area taking into account strictly defined criteria. The findings not only present the current state of research in this field, but may also point to new areas of research. I make some specific comments below.
5 (It seems to me that the second e-mail address should be removed as it refers to a different person.)
25-26 List… discipline (Please remove this sentence.)
148 by by > by
148 GG, WB and MS (Please explain the authors’ initials as they occur for the first time in the text or replace by the expression “the authors of this article” in line 148 and “the authors” in line 150.)
157-158 (Point number 9 is missing)
Table 1 (It appears that the items in each category have been arranged by decreasing years, such information should be given in the text when quoting this table. Please review the table again carefully, as there are some small mistakes in the table, but it is not possible to indicate them exactly, as the rows are not numbered. I also believe the font and line spacing should be reduced.)
459-460 Please… explanation (Please remove this sentence.)
References (References should give the surnames and first letters of the first names of all authors. Please also adapt the notation of references to other specific requirements of the Journal.)
Weakness:
- lack of a precise indication of what the purpose of the research was (the word 'purpose' should be used explicitly in the sentence ending the 'Introduction' section)
- no naming of the research method (comparative analysis of scientific articles)
- a lack of information as to why the Scopus database was searched twice in such a short interval (did this significantly affect the results?)
- Table 1 is too large (it could be divided into four smaller tables, depending on the areas indicated by the authors)
- inadequacy of References to meet editing requirements
Author Response
|
Comments |
Response |
|
|
Reviewer 1 |
5 (It seems to me that the second e-mail address should be removed as it refers to a different person.) |
Now removed as suggested |
|
25-26 List… discipline (Please remove this sentence.) |
Now removed |
|
|
148 by by > by |
Now amended |
|
|
148 GG, WB and MS (Please explain the authors’ initials as they occur for the first time in the text or replace by the expression “the authors of this article” in line 148 and “the authors” in line 150.) |
Updated as suggested |
|
|
157-158 (Point number 9 is missing) |
Now amended |
|
|
Table 1 (It appears that the items in each category have been arranged by decreasing years, such information should be given in the text when quoting this table. Please review the table again carefully, as there are some small mistakes in the table, but it is not possible to indicate them exactly, as the rows are not numbered. I also believe the font and line spacing should be reduced.) |
We have now included in the text that the table is arranged in descending order. Table also checked for small mistakes |
|
|
459-460 Please… explanation (Please remove this sentence.) |
Now removed |
|
|
References (References should give the surnames and first letters of the first names of all authors. Please also adapt the notation of references to other specific requirements of the Journal.) |
References now formatted using MDPI requirements |
|
|
- lack of a precise indication of what the purpose of the research was (the word 'purpose' should be used explicitly in the sentence ending the 'Introduction' section) |
We have now been more explicit about what the purpose of the scoping review is. |
|
|
- no naming of the research method (comparative analysis of scientific articles) |
Information on the research method now included in section 2.5. We used a narrative review / descriptive-analytical approach to chart the data, as recommended by Arksey and O’Malley (2005) |
|
|
- a lack of information as to why the Scopus database was searched twice in such a short interval (did this significantly affect the results?) |
Explanation now added. The database was searched twice as the authors were aware of new papers being published in the months leading up to submission or article |
|
|
- Table 1 is too large (it could be divided into four smaller tables, depending on the areas indicated by the authors) |
Table now divided into four smaller tables as suggested |
|
|
- inadequacy of References to meet editing requirements |
References and citations now formatted according to journal requirements |
Reviewer 2 Report
The authors retrieved 23 articles relating to efforts for sustainability at school from SCOPUS and Google. These articles were classified into 4 categories; 1) interventions for sustainable school menus; 2) school food waste reduction; 3) sustainable food system education through the use of school gardens; and 4) dietary interventions with added environmental components. The authors suggested from this scoping review a new direction of research including a long follow-up study, a controlled study, and utilization of school gardens. This preliminary article for the future research can also provide advice to journal readers. But there are a few concerns.
Major points
1. The Discussion includes repetition of the results, and the authors’ thought with few references, not based on evidence. Ref. 42-47 were cited in the Discussion. The authors wrote just only that the references did not mention sustainability. Most of them did not support the ideas. References should be added from new viewpoints, for example, efforts for other age-groups or in office lunch, other effects of experience in gardening, cooking or other activity. The Discussion should be revised thoroughly.
Minor points
2. The manuscript should be deliberately prepared.
1) L23 in the Abstract “Further research is needed to explore the effectiveness of including ‘environmental sustainability’ as a primary outcome of such interventions.” This dose not reflected the conclusion.
2) L165–171 has 23 references from Ref. 17 to 39. Ref. 40 appears in L173. Table 1 has 23 references from Ref. 17 to 40 excluding Ref. 24. Are there 24 references?
3) Table 1. “Author, year and location” item. The location of affiliation is not interesting. The country in which schools exist is interesting.
4) L343–345. “We are aware that other interventions may have been developed / are currently in use, but the literature regarding evaluation of these does not appear to be keeping pace.” This was not mentioned in the Results.
3. The authors may be native speaker of English, but this review is not native. I might be wrong, but the manuscript seems wordy and to have grammatical errors and not to comply with the journal instructions.
1) Table 1 may be formatted with the template. But it is not readable. Font size should be smaller. Paragraphs are left-aligned. “To facilitate the copy-editing of larger tables, smaller fonts may be used, but no less than 8 pt. in size. “
Colombo et al. 2021 [25] “kitchen staff f” what is “f”
Perez-Neira et al. 2021 [18] “3) when and how the products are consumed 4) what products are consumed,” starts with uppercase as 1) and 2)
CO2 unit is not consistent.
Battle-Bayer et al. 2021 [20] Serif and no-Serif fonts are mixed.
Elinder et al. 2020 [26] itemization is broken.
Martinez et al. 2020 [19] itemization is broken.
Lombardini et al. 2013 [32] What is “no s.d between groups”?
Lochner et al. 2021 [40] country names should be displayed.
Cramer et al 2019 [34] uppercase N/A and lowercase n/a are mixed.
Goldberg et al. 2015 [35] What is “no s.d between groups”?
2) L157 “7) comparator(s);” a single comparator is odd. Groups, subgroups, etc.
L211 “astringent menu” should be explained.
L233 and other text. All references must be numbered in order of appearance in the text, and all reference numbers should be placed in square brackets [ ]. L223 “Elinder et al. 223 (2020) and Colombo et al. (2020)” L228, L230, L240, L246,,,,,etc
L254, and other text and Table1. A space is inserted before unit “g”.
L281 “starter” should be explained.
L371 the second brackets are not necessary.
L406 no refefences.
L439 (ref) references are forgotten.
L457 quotation marks and a sentence after the quotation (L459) are not needed.
Several punctuations are forgotten. For example, “27] three tested” L222.
The conclusion should be written concisely.
3) References are not formatted complying with the instructions. An issue number and “p.” before a page range are not necessary. The instructions say “DOI numbers (Digital Object Identifier) are not mandatory but highly encouraged. “
Author Response
|
The Discussion includes repetition of the results, and the authors’ thought with few references, not based on evidence. Ref. 42-47 were cited in the Discussion. The authors wrote just only that the references did not mention sustainability. Most of them did not support the ideas. References should be added from new viewpoints, for example, efforts for other age-groups or in office lunch, other effects of experience in gardening, cooking or other activity. The Discussion should be revised thoroughly. |
We have now revised the discussion to include evidence from other studies and viewpoints to support ideas |
|
L23 in the Abstract “Further research is needed to explore the effectiveness of including ‘environmental sustainability’ as a primary outcome of such interventions.” This dose not reflected the conclusion. |
This statement has now been updated. |
|
L165–171 has 23 references from Ref. 17 to 39. Ref. 40 appears in L173. Table 1 has 23 references from Ref. 17 to 40 excluding Ref. 24. Are there 24 references? |
References have now been checked and amended throughout |
|
Table 1. “Author, year and location” item. The location of affiliation is not interesting. The country in which schools exist is interesting. |
The country in which schools are located have now been added to relevant column |
|
L343–345. “We are aware that other interventions may have been developed / are currently in use, but the literature regarding evaluation of these does not appear to be keeping pace.” This was not mentioned in the Results. |
We have now moved this statement to our strengths and limitations section. We included this to acknowledge that school based interventions aimed at improving environmental sustainability of the school food system may have been tested locally or outside of academia, but that the literature on these may not be available, and therefore they may not be represented in the review. |
|
1)Table 1 may be formatted with the template. But it is not readable. Font size should be smaller. Paragraphs are left-aligned. “To facilitate the copy-editing of larger tables, smaller fonts may be used, but no less than 8 pt. in size. 2)Colombo et al. 2021 [25] “kitchen staff f” what is “f” 3) Perez-Neira et al. 2021 [18] “3) when and how the products are consumed 4) what products are consumed,” starts with uppercase as 1) and 2) 4) CO2 unit is not consistent. 5) Battle-Bayer et al. 2021 [20] Serif and no-Serif fonts are mixed. 6) Elinder et al. 2020 [26] itemization is broken. Martinez et al. 2020 [19] itemization is broken. 7) Lombardini et al. 2013 [32] What is “no s.d between groups”? 8) Lochner et al. 2021 [40] country names should be displayed. 9) Cramer et al 2019 [34] uppercase N/A and lowercase n/a are mixed. 10) Goldberg et al. 2015 [35] What is “no s.d between groups”? 11) L157 “7) comparator(s);” a single comparator is odd. Groups, subgroups, etc. 12) L211 “astringent menu” should be explained. 13) L233 and other text. All references must be numbered in order of appearance in the text, and all reference numbers should be placed in square brackets [ ]. L223 “Elinder et al. 223 (2020) and Colombo et al. (2020)” L228, L230, L240, L246,,,,,etc 14) L254, and other text and Table1. A space is inserted before unit “g” 15) L281 “starter” should be explained. 16) L371 the second brackets are not necessary. 17) L406 no references. 18) L439 (ref) references are forgotten. 19) L457 quotation marks and a sentence after the quotation (L459) are not needed. 20) Several punctuations are forgotten. For example, “27] three tested” L222. |
1) Table now re-formatted as suggested 2) ‘f’ now removed 3) Upper case now used 4) We have used the CO2 unit of measurement as detailed in each paper - these do differ between studies 5) Now amended 6) The references have been numbered by the order they appear in the article, therefore, the ordering of the numbers do not flow consistently in the table. We welcome suggestions on how to avoid this. 7) s.d. (significant difference) has now been spelt out in full
8) Countries now displayed 9) All now upper case N/A 10) Now spelt out in full 11) Now updated to comparator group) 12) Explanation now provided in table when describing intervention 13) We have used endnote software to manage our citations which automatically numbers the citations in order of appearance. The order may not be consistent in some sections of the article of the citations that have appeared earlier in the paper (e.g., citations in section 3.1 have already appeared in section 3.0 when summarising characteristics of the papers, and therefore appear in a different order here). 14) Space now added where previously missing 15) Explanation now provided in table when describing intervention 16) now removed 17) now added 18) now added 19) now removed 20) amended |
|
The conclusion should be written concisely. |
We have now reworded the conclusion to be more concise |
|
References are not formatted complying with the instructions. An issue number and “p.” before a page range are not necessary. The instructions say “DOI numbers (Digital Object Identifier) are not mandatory but highly encouraged. “ |
References now formatted according to MDPI requirements |
Reviewer 3 Report
This review explored interventions aiming to improve the environmental sustainability of school food systems with a scoping review methodology.While a considerable body of literature focuses on school food intervention studies that focus on improving children's dietary health, there is little literature exploring sustainable school food systems with environmental outcomes as the primary goal, and some schools and communities are using local initiatives to improve the sustainability and environmental impact of school food, research in this area does not appear to have kept up pace.Thus, after screening 6,016 records for eligibility; 23 of which were eligible for inclusion in the scoping review, the resulting findings provide useful knowledge additional to this topic area. It is suggested to revise the study design and discuss findings in the context of existing studies, as follows.
â‘ Please explain the rationale and basis for setting eligibility criteria for records.
â‘¡Please present the validity and consistency checks of the screened records.
â‘¢Please explain what is the purpose of the additional supplementary sample? And how is the validity and consistency of the study ensured?
â‘£3.1 Is the word “sustainability” in the sub-heading accurate and appropriate? It is ambiguous because the studies mentioned show mixed results for menus with reduced meat consumption, so the sustainability of the menu is not known. Also, in terms of low consumption, the menu elements in 3.2 designed to reduce food waste could be classified as sustainable to some extent.
⑤The implementation areas for the screened records are mainly developed countries such as Europe and the USA, why is this not reflected in the discussion?
â‘¥Please add the internal logical relationship between the four most common interventions.
Title:
It’s all right.
Abstract:
The content of the background needs to be reduced, and the specific review methods should be mentioned in the method.
Introduction:
The main object of this study is the environmental sustainability of school food systems, but the first and second paragraphs mainly describe the significance of the environmental sustainability of food systems, without providing more robust and practical data and examples related to school food systems
Methods:
â‘ Please add the innovation of the scope limitation method.
â‘¡Can the part of the “Search terms” be merged with the section of “Identifying relevant studies”?
â‘¢Can you elaborate on the relevant stakeholders mentioned in the context section of the study selection?
Results:
①In the second paragraph, why does it only mention that there are 15 records instead of 23?
②Why validity and consistency tests are not represented in Table 1?
â‘¢Please delete the redundant "f" in the “Intervention group” section of Table 2 of 29.
â‘£Please add a separator for the numbers over 1,000, e.g. 1635 in in the “Intervention group” section of Table 4 of 29 should be 1,635. Check all numbers including those in the tables.
⑤Please check all format, e.g. CO2-eq in “Outcome(s) related to sustainable school food systems” section of Table 6 of 29 should be in subscript.
â‘¥Please check the integrity of the brackets, e.g. “(“ in “Intervention” section of Table 7 of 29 .
⑦Please write in uniform letters, e.g. “N/A”or “n/a” in “Comparison or control group”section
â‘§Please add the character “n=” before the value”177 “ in “Comparison or control group”section of Table 15 of 29.
Author Response
|
Reviewer 3 |
Please explain the rationale and basis for setting eligibility criteria for records. |
We followed the Arskey and O’Malley (2005) methodology for undertaking scoping reviews. They advise setting eligibility criteria that enable records to be eliminated that do not address the research question. Our eligibility criteria were selected based upon the study concept and study setting and to ensure records were identified from appropriate evidence sources. Section 2.4 has now been updated to justify the rationale for setting eligibility criteria |
|
Please present the validity and consistency checks of the screened records. |
Validity checks were not statistically assessed (as this is not a requirement of the methodology we were following). Rather, reviewers sat down together periodically throughout the review to discuss uncertainties around eligibility until agreement and clarity was reached. We have now altered the text to avoid wording which suggests that formal validity and consistency checks did take place |
|
|
Please explain what is the purpose of the additional supplementary sample? And how is the validity and consistency of the study ensured? |
The purpose of including papers that were identified through citations of included papers was to ensure some representation of papers that were not identified from the scoping search. Papers identified in this way still needed to meet eligibility criteria as agreed by WB, MS and GG. Section 2.2 has now been updated to reflect this |
|
|
Is the word “sustainability” in the sub-heading accurate and appropriate? It is ambiguous because the studies mentioned show mixed results for menus with reduced meat consumption, so the sustainability of the menu is not known. Also, in terms of low consumption, the menu elements in 3.2 designed to reduce food waste could be classified as sustainable to some extent. |
We agree this could be misleading. We have now changed this to ‘school menus designed to be more sustainable throughout’ |
|
|
The implementation areas for the screened records are mainly developed countries such as Europe and the USA, why is this not reflected in the discussion? |
Whilst of interest, we feel that the differences between these countries and others (context, environment, policy) makes it difficult to draw conclusions and recommendations based on the data we have available from this review |
|
|
Please add the internal logical relationship between the four most common interventions. |
We have discussed this point amongst our team and are not clear on what is being requested here? If we are being asked to discuss commonalities between the interventions, this would be that they are all delivered within a school context, but this was expected given the scope of the review. |
|
|
Abstract: The content of the background needs to be reduced, and the specific review methods should be mentioned in the method.
|
The abstract has now been updated as suggested |
|
|
Introduction: The main object of this study is the environmental sustainability of school food systems, but the first and second paragraphs mainly describe the significance of the environmental sustainability of food systems, without providing more robust and practical data and examples related to school food systems |
We have reworked the introduction to make it more relevant to the school food system context |
|
|
Methods: 1) Please add the innovation of the scope limitation method. 2) Can the part of the “Search terms” be merged with the section of “Identifying relevant studies”? - 3) Can you elaborate on the relevant stakeholders mentioned in the context section of the study selection? |
1) We have added more detail on our exclusion criteria as requested. 2) We have now merged these sections as suggested 3) More detail now added on stakeholders mentioned in the context section |
|
|
Results: 1) In the second paragraph, why does it only mention that there are 15 records instead of 23? 2) Why validity and consistency tests are not represented in Table 1? 3) Please delete the redundant "f" in the “Intervention group” section of Table 2 of 29. 4) Please add a separator for the numbers over 1,000, e.g. 1635 in in the “Intervention group” section of Table 4 of 29 should be 1,635. Check all numbers including those in the tables. 5) Please check all format, e.g. CO2-eq in “Outcome(s) related to sustainable school food systems” section of Table 6 of 29 should be in subscript. 6) Please check the integrity of the brackets, e.g. “(“ in “Intervention” section of Table 7 of 29 . 7) Please write in uniform letters, e.g. “N/A”or “n/a” in “Comparison or control group”section 8) Please add the character “n=” before the value”177 “ in “Comparison or control group”section of Table 15 of 29. |
1) We now describe he study design of all studies within this paragraph 2) As above, formal validity and consistency tests were not undertaken as part of this scoping review. Wording in section 2.3 has now been updated to avoid confusion. 3) “f” now removed 4) Punctuation errors now corrected in table 5) CO2 now expressed correctly 6) Integrity of brackets now corrected 7) N/A now expressed consistently 8) “n” now entered |
Round 2
Reviewer 2 Report
Reference numbers remain in disorder and are confused although the authors amended them. It is not understandable whether the context of the reference matches the results. “resulting in 23 studies/reports …(Figure 1)” in L276. Tables 1–4 have 13, 6,3, and 2, articles, respectively, which totals 24 articles. Ref. 46 appears twice in Table 3. Ref. 41 and 42 in L279 do not appear in Tables. “3.1. Food waste reduction Six of the studies identified in the review aimed to reduce the amount of food wastedfrom school lunches [30-32,37,47,51].” Ref. 51 is inappropriate here. Figure 1 should be corrected.
|
Table number |
Row number |
Ref. number |
|
Table 1 |
1 |
36 |
|
Table 1 |
2 |
33 |
|
Table 1 |
3 |
27 |
|
Table 1 |
4 |
29 |
|
Table 1 |
5 |
34 |
|
Table 1 |
6 |
45 |
|
Table 1 |
7 |
35 |
|
Table 1 |
8 |
28 |
|
Table 1 |
9 |
44 |
|
Table 1 |
10 |
26 |
|
Table 1 |
11 |
48 |
|
Table 1 |
12 |
40 |
|
Table 1 |
13 |
38 |
|
Table 2 |
1 |
47 |
|
Table 2 |
2 |
31 |
|
Table 2 |
3 |
32 |
|
Table 2 |
4 |
37 |
|
Table 2 |
5 |
30 |
|
Table 2 |
6 |
50 |
|
Table 3 |
1 |
49 |
|
Table 3 |
2 |
46 |
|
Table 3 |
3 |
46 |
|
Table 4 |
1 |
43 |
|
Table 4 |
2 |
39 |
Section numbers in Results 3 also fell in disorder. Section number 3.1 is repeated.
Could you add school lunch style in each country in Tables if available? Cafeteria type (selecting foods freely; eg. US?), selecting two or three options of meals, or providing the meal at each school (eg. Japan).
L56–58 “Children spend a large proportion of their time at school, and an average of 30% of children’s daily energy intake is suggested to come from school food [4].” WFP reported coverage of school meals is 41%. https://www.wfp.org/publications/state-school-feeding-worldwide-2022. Is it overestimated?
WFP uses SDG as multiple Sustainable Development Goals: address poverty (SDG1), hunger (SDG2), health and wellbeing (SDG3), education (SDG4), gender equality (SDG5), economic growth (SDG8), reduced inequality (SDG10) and global partnerships (SDG17). https://www.wfp.org/publications/changing-lives-school-based-programmes. Can “Environmental sustainability” be strengthened in the Title, Abstract, and Introduction? https://www.wfp.org/school-feeding
L44 “Member States of the United Nations” “within University students” in Discussion. Upper case is needed?

Author Response
|
1) Reference numbers remain in disorder and are confused although the authors amended them. It is not understandable whether the context of the reference matches the results. “resulting in 23 studies/reports …(Figure 1)” in L276. Tables 1–4 have 13, 6,3, and 2, articles, respectively, which totals 24 articles. 2) Ref. 46 appears twice in Table 3. 3) Ref. 41 and 42 in L279 do not appear in Tables. 4) “3.1. Food waste reduction Six of the studies identified in the review aimed to reduce the amount of food wasted from school lunches [30-32,37,47,51].” Ref. 51 is inappropriate here. Figure 1 should be corrected. |
1) Thank you for bringing this to our attention, we have now amended figure 1
2) amended 3) amended 4) amended - please note this was a citation error. Ref 51 is not one of the included papers, therefore does not need amending in figure 1 |
|
Section numbers in Results 3 also fell in disorder. Section number 3.1 is repeated. |
amended |
|
Could you add school lunch style in each country in Tables if available? Cafeteria type (selecting foods freely; eg. US?), selecting two or three options of meals, or providing the meal at each school (eg. Japan). |
We have now provided more context in the sample characteristics column where available |
|
L56–58 “Children spend a large proportion of their time at school, and an average of 30% of children’s daily energy intake is suggested to come from school food [4].” WFP reported coverage of school meals is 41%. https://www.wfp.org/publications/state-school-feeding-worldwide-2022. Is it overestimated? |
The reference you are referring to suggests 41% of children receive a school meal. Our reference refers to data which suggests 30% of a child’s daily dietary intake comes from school food |
|
1) WFP uses SDG as multiple Sustainable Development Goals: address poverty (SDG1), hunger (SDG2), health and wellbeing (SDG3), education (SDG4), gender equality (SDG5), economic growth (SDG8), reduced inequality (SDG10) and global partnerships (SDG17). https://www.wfp.org/publications/changing-lives-school-based-programmes. 2) Can “Environmental sustainability” be strengthened in the Title, Abstract, and Introduction? https://www.wfp.org/school-feeding |
1) updated to reflect this 2) We are not quite clear what this comment means but have now referred to “strengthening” environmental sustainability throughout |
|
L44 “Member States of the United Nations” “within University students” in Discussion. Upper case is needed? |
“Member States of the United Nations” is capitalised by United Nations in all literature, which is who we have cited here in the introduction |